# Infants’ Vitamin D Nutritional Status in the First Year of Life in Northern Taiwan

**DOI:** 10.3390/nu12020404

**Published:** 2020-02-04

**Authors:** Chiao-Ming Chen, Shu-Ci Mu, Yi-Ling Chen, Li-Yi Tsai, Yung-Ting Kuo, In-Mei Cheong, Mei-Ling Chang, Sing-Chung Li

**Affiliations:** 1Department of food Science, Nutrition, and Nutraceutical Biotechnology, Shih Chien University, Taipei 10462, Taiwan; charming@g2.usc.edu.tw (C.-M.C.); mlchang@g2.usc.edu.tw (M.-L.C.); 2School of Medicine, Fu-Jen Catholic University, New Taipei City 24205, Taiwan; musc1006@gmail.com; 3Department of Pediatrics, Shin-Kong Wu Ho-Su Memorial Hospital, Taipei 11101, Taiwan; Ylchen1219@gmail.com (Y.-L.C.); Ly.tsai95@gmail.com (L.-Y.T.); 4School of Medicine, Taipei Medical University, Taipei 11031, Taiwan; 5Institute of Environmental & Occupational Health Sciences, College of Public Health, National Taiwan University, Taipei 10617, Taiwan; 6Department of Pediatrics, Shuang Ho Hospital, Ministry of Health and Welfare, Taipei Medical University, New Taipei 23561, Taiwan; pedkuoyt@tmu.edu.tw; 7Department of Pediatrics, School of Medicine, College of Medicine, Taipei Medical University, Taipei 11031, Taiwan; 8School of Nutrition and Health Sciences, College of Nutrition, Taipei Medical University, 250 Wu-Hsing Street, Taipei 11031, Taiwan

**Keywords:** infant, breastfeeding, formula, vitamin D deficiency

## Abstract

Vitamin D deficiency (VDD) and insufficiency (VDI) are common among exclusively breastfeeding infants. However, epidemiological evidence for the prevalence of VDD in infants during their first year of life in Taiwan has never been found. This trial determined the prevalence of VDD and VDI and the association between dietary vitamin D and vitamin D nutritional status in Northern Taiwan. A cross-sectional study was conducted on infants who returned to well-baby examinations from October 2012 to January 2014 in three hospitals: Shin Kong Wu Ho-Su Memorial Hospital, Taipei Medical University Hospital, and Shuang Ho Hospital. The specific vitamin D cut-off concentrations for VDD, VDI, and VDS are 25(OH)D_3_ levels ≤20, 21–29, and ≥30 (ng/mL). Overall, 481 infants’ parents completed a questionnaire comprising questions related to vitamin D nutritional status, including weekly time outdoors, breastfeeding status, anthropometric measurement, and assessment of dietary intake, including milk and complementary food. The results revealed that 197 (41%) and 212 (44%) of infants in their first year of life had VDI and VDD, respectively, by the Endocrine Society guidelines. Breastfed infants had a higher prevalence of VDI (86.1%) than did mixed-fed (51.9%) and formula-fed (38.5%) infants (*p* < 0.001). The prevalence of VDD was 55.4% in infants aged under six months but increased to 61.6% in infants aged over six months. Infants in the VDI and VDD groups had the same anthropometrics as those in the vitamin D sufficiency (VDS) group. Our results revealed that 25(OH)D_3_ had a negative correlation with the intact parathyroid hormone (iPTH) when the serum 25(OH)D_3_ level ≤20 ng/mL (r = −0.21, *p* = 0.001). The VDS group had a higher total vitamin D intake than did the VDI and VDD groups, which was mainly obtained from infant formula. Our data revealed that dietary vitamin D intake and birth season were major indicators in predicting VDD. Lower dietary vitamin D intake and born in winter and spring significantly increased the odds ratio (OR) for VDI by 1.15 (95% CI 1.09–1.20) and 2.02 (95% CI 1.10–3.70), respectively, and that for VDD by 1.23 (95% CI 1.16–1.31) and 2.37 (95% CI 1.35–4.17) without covariates adjustment, respectively. Furthermore, ORs for VDI and VDD significantly differed after adjustment for covariates. In conclusion, the prevalence of VDI and VDD were high in infants during the first year of life. Breastfeeding infants had difficulty in obtaining sufficient vitamin D from diet. In cases where the amount of sun exposure that is safe and sufficient to improve vitamin D status is unclear, breastfed infants aged below one year old are recommended to be supplemented with vitamin D.

## 1. Introduction

Vitamin D is a fat-soluble vitamin that generally refers to two prohormones: ergocalciferol (vitamin D2) and cholecalciferol (vitamin D3). Vitamin D2 is largely human-made and added to food, whereas vitamin D3 is synthesized in the skin of humans from 7-dehydrocholesterol and is also consumed in the diet through animal-based foods [1,2]. Vitamin D is first converted to 25-hydroxyvitamin D (25(OH)D), also known as calcidiol, in the liver. It then appears primarily in the kidney, then known as calcitriol, and forms physiologically active 1.25-dihydroxyvitamin D (1.25(OH)2D) [3]. 

The American Academy of Pediatrics (AAP) recommends that exclusively and partially breastfed infants should supply 400 IU/day of vitamin D after birth and continue to receive vitamin D supplements until they are weaned and consume more than 1000 mL/day of vitamin D-fortified formula milk [4]. In France, infant consumption of oral vitamin D supplements not exceeding 1000 IU/day through vitamin D-fortified milk does not appear to induce vitamin D overloading during the first three months of life [5]. However, vitamin D intoxication appears to be caused by excessive vitamin D3 fortification in dairy milk [6].

In the human body, both vitamin D2 and D3 are combined with vitamin D-binding proteins in the blood and transported to the liver. 25(OH)D_3_ is the major circulating form of vitamin D and provides the single most efficient assessment of vitamin D nutritional status in infants [7,8]. 25(OH)D_3_ activates vitamin D receptors and circulates in human blood at approximately 1000 times the concentration of (1.25(OH)2D) [9]. In addition, for either isoform of vitamin D, infant blood 25(OH)D_3_ concentration does not differ between low-dose supplementations of vitamin D2 or D3 in breastfed infants (one month old) for three months [10]. Vitamin D and calcium deficiency in infants can lead to bone malformation, seizures, and difficulty breathing. Severe vitamin D deficiency (VDD) with 25(OH)D_3_ levels less than 10 ng/mL leads to overt skeletal abnormalities in children, typically defined as rickets [11]. However, numerous infants, children, and adolescents that have vitamin D insufficiency (VDI) have no apparent skeletal or calcium metabolism abnormalities [12].

Breastmilk is the natural first food for infants, is considered the best source of nutrition for normal newborns in the first months of life, and provides optimal nutrition and health protection for the first six months, after which breastmilk is continued along with complementary foods from six to 12 months for nearly all infants [13]. However, the milk of healthy lactating women contains a low vitamin D concentration, leading to a poor source of vitamin D for the exclusive breastfeeding [14,15,16].

Studies have indicated that vitamin D nutritional status affects neonatal anthropometrics, bone development, electrolyte balance, immune function, and cognitive function [17,18,19]. Exclusively breastfed infants depend on sunlight exposure and vitamin D intake from breastmilk. However, studies have indicated that the vitamin D content in breastmilk is low, which increases the risk of VDD in infants living in areas with insufficient sunlight [20,21]. Taiwan is in the low latitudes and has sunny areas; thus, infants are considered to have fewer problems with VDD. However, the prevalence of VDD and VDI, as well as the factors affecting the vitamin D nutritional status of infants under one year old, remains unclear. This study investigated the vitamin D status of infants in their first year of life and its associated factors.

## 2. Participants and Methods

### 2.1. Study Participants

This study was part of the survey “The effects of breastfeeding on iron and vitamin D status in infants”. A cross-sectional study was conducted in three hospitals: Shin Kong Wu Ho-Su Memorial Hospital, Taipei Medical University Hospital, and Shuang Ho Hospital from October 2012 to January 2014. Overall, 2804 healthy infants who came to clinic for well-baby examinations were screened for eligibility. The inclusion criteria for infants and mothers enrolled were as follows: no systemic diseases (toxemia, hypertension, diabetes mellitus, and heart disease) during pregnancy; infant age below one year; and good health and no diseases, as determined by a pediatrician. Infants with premature birth; congenital diseases (such as heart, lung, liver, and intestinal diseases); growth disorders; diagnosis of gastrointestinal disorders (nausea, vomiting, pain, flatulence, diarrhea, and malabsorption); and thalassemia were excluded. 

The Ethics Committee of Shin Kong Wu Ho-Su Memorial Hospital and Taipei Medical University approved this study, in accordance with the International Ethical Guidelines for Biomedical Research Involving Human Subjects and ethical principles of the Declaration of Helsinki (20120901R and 201308004). Written informed consent was obtained from all participants or legal representatives before the study procedures were performed.

### 2.2. Basic Characteristics and Dietary Vitamin D Intake Assessment 

The infants’ parents completed modified and validated questionnaires on their demographic and dietary information. Infant basic information such as gestational week, chronological age, and birth weight were recorded per the medical chart. Body length and weight were measured using an infantometer and weighing scales. Head circumference was measured by applying a plastic tape around the forehead (above the eyebrows) and the occipital protuberance. These measurements were converted into percentiles according to the growth charts for Taiwanese children released by the Ministry of Health and Welfare. Diet was assessed using a semi-quantitative frequency method, and food items included tofu, chicken, pork, beef, fish, egg yolks, viscera, vegetables, rice, wheat, baby rice cereal, baby wheat cereal, formula milk, juice, and purees. Complementary food and formula milk intake were quantified from standard bowls, spoons, and feeding bottles. Food vitamin D data are not available in the Taiwanese Food Composition Table. Lee et al. consulted several sources to compile an in-house vitamin D food composition database [22]. Those included: the fourth version of Standard Tables of Food Composition in Japan, the fifth version of Standard Tables of Food Composition in Japan, Food Values of Portions Commonly Used, USDA NUTRIENT DATA LABORATORY, and Finnish Food Composition Database. Vitamin D intake from complementary food was calculated using this integrated database. In addition, vitamin D intake from formula milk and baby cereal were calculated according to their nutrient labels. The vitamin D content of breastmilk is reported to be minor [23] and was not evaluated in this study. Exclusive breastfeeding months, birth season, and weekly time outdoors were reported by parents.

### 2.3. Blood Collection

Blood samples (4 mL) were collected from an arterial puncture of one arm into vacutainers without anticoagulant by a trained technician. Serum was collected after centrifugation at 1400 *g* for 10 min at 4 °C and immediately sent to the Central Laboratory, Shin Kong Wu Ho-Su Memorial Hospital for blood analysis. 

### 2.4. Blood Measurements

Intact parathyroid hormone (iPTH) was measured with the iPTH enzyme-linked immunosorbent assay kit, which applied a competitive enzyme immunoassay technique with a monoclonal anti-iPTH antibody and an iPTH-horseradish peroxidase conjugate from My BioSource (San Diego, CA, USA). The assay sample and buffer were incubated together with the iPTH-HRP conjugate in a precoated plate for 1 h. After incubation, the wells were decanted and washed 5 times. The wells were then incubated with a substrate for HRP enzymes. The product of the enzyme–substrate reaction formed a blue complex. Finally, a stop solution was added to stop the reaction, after which the solution turned yellow. The color intensity was measured spectrophotometrically at 450 nm in a microplate reader. Intra- and inter-assay coefficients of variation (CV) were 1.2% and 0.9%, respectively.

### 2.5. Blood Vitamin D Analyses through Ultraperformance Liquid Chromatography–Electrospray Ionization–Mass Spectrometry (UPLC-ESI-MS)

A total of 0.5 mL of serum was added to 200 uL (100 ng/mL) of 1α-hydroxyvitamin D_3_ (Sigma Chemical Co., St. Louis, MO, USA) as the internal standard and then added 2 mL of ethanol in a sample tube. The tubes were mixed in a multitube vortex mixer for 30 s and 25(OH)D_3_ was extracted after the tubes were mixed 2 times (60 s each) with 3 mL of hexane. The phases were separated through centrifugation, and the upper organic phase was transferred to a conical tube and evaporated with a ScanVac CoolSafe™ ScanSpeed MaxiVac centrifuge for vacuum evaporation (Lynge, Denmark). The remaining dry matter was dissolved in 100 μL of methanol and identified through UPLC-ESI-MS. 

The liquid chromatography-ESI-MS system consisted of a UPLC system (Ultimate 3000 RSLC, Dionex, Idstein, Germany) and an atmospheric pressure chemical ionization (APCI) source quadrupole time-of-flight mass spectrometer (maXis ultra-high resolution (UHR-qToF) system (Bruker Daltonics, Bremen, Germany). The autosampler was set to 4 °C. Separation was performed through reversed-phase liquid chromatography on a BEH C18 column (2.1 × 100 mm, Waters, Massachusetts, USA). Elution was carried out with 25% mobile phase A (2 mM ammonium formate in ultrapure water) and 75% mobile phase B (2 mM ammonium formate in methanol), held at 75% B for 2 min, raised to 80% B in 1 min, further raised to 100% B in 4.5 min, held at 100% B for 1.5 min, and then lowered to 75% B in 1 min. The column was equilibrated by pumping 75% B for 4 min. The flow rate was set to 0.3 mL/min. LC-APCI-MS chromatograms were acquired in the positive ion mode under the following conditions: capillary voltage of −1500 V, endplate offset of −500 V, corona of +5000 nA, vaporized temperature of 400 °C, dry temperature at 180 °C, dry gas flow maintained at 5 L/min, nebulizer gas at 1.6 bar, and acquisition range of m/z 50–1000. Retention times of 25(OH)D_3_ and 1α-hydroxyvitamin D_3_ were 6.8 min and 8.7 min. The extract standard for calibration curves was constructed using five concentrations of 25(OH)D_3_ (Sigma Chemical Co., St. Louis, MO, USA), r = 0.99. The recovery rate was > 90%, and the detection limit was 10 nM. Intra- and inter-assay CV were 1.0% and 5.0%, respectively. All the listed experiments and procedures were performed by a technician in the Technology Commons, College of Life Science, National Taiwan University. 

### 2.6. Statistical Analysis

The Endocrine Society guidelines defines VDD as having a serum 25(OH)D_3_ level less than 20 ng/mL, a serum 25(OH)D_3_ level between 21–29 ng/mL for VDI, and a serum 25(OH)D_3_ level over 30 ng/mL for vitamin D sufficiency (VDS) [24]. We divided the participants into three groups according to these definitions: VDD, VDI, and VDS groups. All data were confirmed to have a normal distribution through the Kolmogorov–Smirnov test. Data were presented as means ± standard deviations, median (interquartile range), or percentage. Intergroup differences were determined using one-way ANOVA, followed by the Scheffé method for post hoc tests or nonparametric statistics. Pearson’s chi-squared test was used to assess categorical variables. The correlation between serum 25(OH)D_3_ level and iPTH was determined through the Pearson correlation test. The association between dietary vitamin D intake and vitamin D nutritional status was determined through multivariable logistic regression. All data analyses were performed with SPSS (v19; SPSS Inc., Chicago, IL, USA). Differences were considered significant at *p* < 0.05.

## 3. Results 

### 3.1. Participant Characteristics

A total of 2804 infants were screened from three hospitals for eligibility in this trial. After exclusion of infants with premature birth, multiple births, congenital diseases, growth diseases, gastrointestinal disorders, and thalassemia, 1368 infants were eligible for this study. However, 779 mothers did not provide consent to extract their infants’ blood. Therefore, 589 infants were ultimately enrolled in this study. However, blood draws were unsuccessful in 39 infants, and not enough blood was drawn to determine serum 25(OH)D_3_ in 28 infants. Thus, a total of 522 cases were included for data analysis. Because not introducing complementary food to infants aged over six months was considered abnormal feeding, four infants aged eight months and two aged 12 months were excluded accordingly. In addition, 35 infants with white blood cell counts >10,000/mm^3^ were suspected of infection and were excluded. Data on the vitamin D status analysis of 481 infants are presented in Figure 1.

According to the Endocrine Society guidelines, 197 (41%) and 212 (44%) infants were diagnosed with VDI and VDD by a physician, respectively (Table 1). No statistically significant intergroup differences were noted in terms of sex, chronological age, gestational age, birthweight, and weekly time outdoors of infants.

The breastfed infants had a significantly higher prevalence of VDD (86.1%) than did mixed-fed (51.9%) and formula-fed (38.5%) infants. Infants with VDD had longer average periods of exclusive breast-feeding (4.2 ± 3.6 months) than did those with VDI (3.1 ± 3.1 months) and VDS (3.1 ± 3.3 months). Infants born during the winter and spring had higher prevalence of VDD (61.8%) than did those born during the summer and fall (52.1%), which was statistically significant.

### 3.2. Vitamin D Status 

Vitamin D statuses are presented in Table 2. Our results showed that the average serum 25(OH)D_3_ level in infants aged 1–12 months was 19.7 ± 9.7 ng/mL, indicating overall vitamin D malnutrition. Although serum 25(OH)D_3_ concentration was significantly low in the VDI and VDD groups, no statistical difference in iPTH was observed. When serum 25(OH)D_3_ level was ≤20 ng/mL, serum 25(OH)D_3_ was negatively correlated with iPTH (r = −0.21, *p* = 0.001; Figure 2A), while serum 25(OH)D_3_ was not correlated with iPTH when serum 25(OH)D_3_ level > 20 ng/mL (r = 0.05, *p* = 0.54). We further divided the data into the following two subgroups: 1–6 and 7–12 months. The data revealed that the correlation between serum 25(OH)D_3_ and iPTH in infants aged 7–12 months was higher than in those aged 1–6 months (r = −0.29, *p* = 0.003 vs. r = −0.23, *p* = 0.005; Figure 2B,C). 

### 3.3. Anthropometrics of Vitamin D Status

Anthropometrics of vitamin D statuses are presented in Table 2. The averages of body weight, body length, and head circumference percentiles were in approximately the 50th percentile. No statistically significant intergroup anthropometric differences were noted, including in terms of body weight, body length, and head circumference percentiles.

### 3.4. Vitamin D Intake of Infants

The total vitamin D intake was calculated as the sum of vitamin D content in formula milk and complementary food, as presented in Table 3. Since complementary food was introduced in only 17 infants and most complementary food comprised cereals and fruit purees for infants aged 1–6 months, the data for vitamin D intake from complementary food is not listed. During ages 1–6 months, there was no difference between groups in chronological age, but the breastfeeding rate was highest in the VDD group. The median daily vitamin D intake from formula milk in the VDS, VDI, and VDD groups were 7.12, 5.33, and 0.00 mg, respectively. Moreover, the VDS group had a higher total vitamin D intake (8.33 mg daily) than did the VDI (5.33 mg daily) and VDD (0.82 mg daily) groups in ages 1–6 months. The lowest total vitamin D intake in infants of the VDD group could be attributed to being breastfed and being fed less formula.

In ages 7–12 months, infants in VDI and VDD groups were significantly older, and the breastfeeding rate was highest in the VDD group. The median daily vitamin D intake from formula milk in the VDS and VDI groups (6.15 mg and 7.70 mg daily, respectively) were significantly higher than in the VDD group (4.10 mg) in ages 7–12 months. This could be attributed to the continuous breastfeeding and less formula fed to infants in the VDD group. No intergroup differences were noted in terms of the vitamin D intake from complementary food. Therefore, formula milk was the major source of vitamin D intake in infants aged 7–12 months. Furthermore, we determined that the prevalence of VDI and VDD in infants aged 1–6 months were 30.4% and 55.4%, respectively, and were 27.3% and 61.6%, respectively, for infants aged 7–12 months.

### 3.5. Determinants of Vitamin D Deficiency among Infants

Logistic regression models were employed to identify predictors of VDI and VDD. In Model 1, no covariates were adjusted. In Model 2, the gestational week, birthweight, sex, body weight percentile, body length percentile, and age of infants were adjusted. Our data revealed that dietary vitamin D intake and birth season were the major indicators in predicting VDI and VDD (Table 4). Regardless of whether the variables were adjusted, VDI and VDD significantly increase the odds ratio (OR) value compared to VDS. The period of exclusive breastfeeding was only associated with VDD. Outdoor times weekly were not associated with vitamin D status.

## 4. Discussion

Serum 25(OH)D_3_ is the best biomarker to assess vitamin D status. It reflects vitamin D produced cutaneously and that obtained from food and supplements [25]. Considering that it is related to bone health, neuroticism, and immune function, the cut points of serum 25(OH)D_3_ concentration have not been developed by a scientific consensus process. The Endocrine Society guidelines defines VDD as having a serum 25(OH)D_3_ level less than 20 ng/mL, a serum 25(OH)D_3_ level between 21–29 ng/mL for VDI, and a serum 25(OH)D_3_ level over 30 ng/mL for VDS [24]. However, the Global Consensus Recommendations define vitamin D deficiency as a serum 25(OH)D_3_ concentration less than 12 ng/mL and insufficiency as 12–20 ng/mL [25]. The Canadian Health Measures Survey suggests 25(OH)D_3_ level less than 12 ng/mL as VDD and those between 12–20 ng/mL as VDI [26]. The prevalence of VDD in infants has been reported to be high in many countries, such as South Korea (49%) [27], China (47%) [28], and India (73%) [29], and low in the United States (12%) [30]. In these studies, VDD was defined as a serum 25(OH)D_3_ level less than 20 ng/mL, except for China (<10 ng/mL). Therefore, the prevalence of VDD in the present study was 44% and similar to Korea, even if our location was on a lower latitude. However, our study revealed that VDD did not affect infants’ body weight, body length, and head circumference in the first year of life. Wierzejska et al. reported no relationship between maternal and neonatal cord blood vitamin D concentrations and neonatal weight, length, head, and chest circumference at birth [31]. However, Song et al. noted that head circumference and birth weight were lower in VDD newborns [28]. Weiler et al. reported that infants with VDD were associated with greater weight and length but lower bone mass relative to body weight [32]. Therefore, a low concentration of vitamin D affects infant skeletal development, and the bone density of these infants must be tracked to understand whether low levels of vitamin D affect infants’ bone health. 

In previous studies, human breastmilk has been reported to be a very poor source of vitamin D and typically contains 5–136 IU/L, even for breastfeeding mothers in an environment with abundant sunlight [23,33]. In addition, breastfeeding women are at higher risk of VDD than nonbreastfeeding women in a German VitaMinFemin study [34]. In our observation, although we did not analyze the vitamin D concentration in breastmilk, breastfeeding infants had a higher incidence of VDD. The vitamin D content in the breastmilk of Taiwanese mothers may also be very low. 

Major and specific risk factors for VDD include low ambient ultraviolet radiation, limited sun exposure, dark skin, obesity, aging, low dietary intake, and gastrointestinal malabsorption [35]. Studies have also discovered VDD in infants’ umbilical cord blood at latitudes of 24°–67° [36]. Taiwan’s latitude is approximately 21°–23°, which is a sunny area. However, recent research has found that VDI in older adults or VDD in healthy individuals is prevalent in subtropical areas, such as Northern Taiwan [37,38]. Xhu et al. revealed that VDD is most severe in late spring and least severe during the summer in preterm births in Northeast China [39]. Low storage of vitamin D in the liver and fat tissue are reasons for low 25(OH)D_3_ levels by seasonal change [40]. In this study, we discovered that infants born during the summer and autumn had better vitamin D nutritional status from being born during the sunniest times of the summer and autumn in Northern Taiwan; however, Northern Taiwan is rainy and cloudy in the spring and winter, which reduces exposure to sunlight.

Vitamin D can be produced through exposure to the sun through the skin. Ultraviolet B (UVB)-mediated production of vitamin D begins when 7-dehydrocholesterol in the skin absorbs UVB radiation (290–315 nm) to generate secosteroid previtamin D3 [41]. Increasing time spent outdoors may improve vitamin D nutritional status [42]. However, in our study, no differences regarding vitamin D nutritional status were observed in infants who were outside each week. The results indicate that Taiwanese infants being outside two times weekly did not exhibit a better vitamin D status. Caregivers should be encouraged to increase the frequency of taking babies outside. Although vitamin D is available through incidental sun exposure, considering the risk of skin damage from sun exposure, the American Academy of Pediatrics recommends avoiding direct, unprotected activities before or after periods of peak sun exposure for newborns and infants [43].

The iPTH regulates fetal–placental mineral homeostasis within the first hours after birth. Parathyroid glands increase the synthesis and secretion of iPTH when concentrations of vitamin D are low, which increases serum Ca, lowers P, stimulates calcitriol synthesis, inhibits calcitriol catabolism, reabsorbs minerals in the kidney tubules, and regulates bone formation [44]. A 25(OH)D_3_ level of 20 ng/mL is generally the threshold of VDD in epidemiologic studies, because serum iPTH tends to increase when the 25(OH)D_3_ concentration is below this value [45]. Our results revealed the serum iPTH level was negatively correlated with the serum 25(OH)D_3_ level when the serum 25(OH)D_3_ level was lower than 20 ng/mL. When the serum 25(OH)D_3_ level was above 20 ng/mL, then no linear correlation between 25(OH)D_3_ and iPTH levels was noted, which was consistent with the results of the study by Martins et al. among Brazilians [46]. Matejek et al. reported a significant relationship between iPTH and 25(OH)D_3_ from the second month of life in healthy preterm newborns and that decreasing the 25(OH)D_3_ level was accompanied by increasing iPTH [47].

In an analysis of dietary vitamin D intake in VDD infants aged 1–6 months, the median vitamin D intake from formula milk was 0, implying that most VDD infants are being breastfed. Taiwan follows the World Health Organization recommendation of exclusively breastfeeding for six months as the optimal approach to feed infants. Thereafter, infants receive complementary food with continued breastfeeding up to two years of age or beyond. Only 17 infants before six months of age were introduced to complementary food, which mostly consisted of cereals and fruit purees with low vitamin D contents. Therefore, the main source of vitamin D for infants before six months of age was formula milk. Although infants over six months of age were provided complementary foods, the vitamin D content from complementary foods was not different among the three groups. Thus, the main source of dietary vitamin D was also formula milk in infants aged 7–12 months. The main sources of vitamin D in complementary food were egg yolks, variety meats, fish, pork, and baby cereal. Baby cereal contributed the most vitamin D content because most were fortified with vitamin D, and the lowest were the variety meats, because most caregivers did not provide variety meats as complementary foods.

Our results revealed that infants with adequate serum 25(OH)D_3_ concentrations had a median vitamin D intake of approximately 8.33–9.41 μg daily. Normal vitamin D stores present at birth are depleted within eight weeks [18]. Infants that were exclusively breastfed before six months of age may have not been able to get sufficient vitamin D because of low vitamin D content in breastmilk; thus, additional vitamin D supplements may have been needed. For infants aged over six months still breastfed, receiving nearly 10 μg (400 IU) of vitamin D from complementary food was difficult. Since the amount of sunlight necessary to produce adequate vitamin D and ensure safety to the baby’s delicate skin remains unclear, vitamin D should be supplemented.

Based on disease-oriented evidence and expert opinions, the AAP recommends that breastfed infants receive 400 IU/day of vitamin D supplements shortly after birth and continue to receive them until they are weaned and consume ≥1000 mL/day of vitamin D–fortified formula or whole milk [4,48]. The Taiwan Pediatric Association revised its guidelines for breastfed infants in 2016 and included the following: (1) Encourage full-term infants to start breastfeeding as soon as possible after birth and (2) continue to breastfeed until one year of age. After one year of age, mothers may continue breastfeeding infants. (3) Exclusively breastfed infants or partially breastfed infants should be provided 400 IU of vitamin D oral supplements daily. Our results revealed that the period of exclusive breastfeeding, birth season, and weekly time outdoors may affect the vitamin D status of infants. With no adjustment of these variables, dietary vitamin D intake was significantly correlated with vitamin D nutritional status. 

The present work was the first to investigate the prevalence of VDI and VDD and discovered that dietary vitamin D intake had a relationship with vitamin D nutrition status in Northern Taiwan. Our results can be used as reference for nutrition policy recommendations. However, some limitations should be considered when interpreting our study results. First, this was a cross-sectional study that was unable to explain the consequence of longitudinal nutritional status on VDI and VDD development. Second, the small sample size and all participants lived in Northern Taiwan, which may limit the generalizability of the results. Therefore, large-sample, multicenter studies are required. Third, although we did not analyze vitamin D concentrations in breastmilk, the average vitamin D content in breastmilk has been very low in previous studies. Forth, due to lack of vitamin D food composition data, we calculated the vitamin D content of complementary foods from various sources. It may not reflect true vitamin D intake. Thus, the vitamin D intake in exclusively breastfed infants was far below the general recommendation of 400 IU per day. In addition, studying the relationship between other nutrients influenced by vitamin D bioavailability and following up on these infants with VDI and VDD are warranted.

## 5. Conclusions

The prevalence of VDI and VDD in infants aged under one year was high in Northern Taiwan, particularly among breastfed infants. Dietary vitamin D intake was the main factor affecting serum 25(OH)D_3_ concentrations in infants. VDD can lead to elevated iPTH, which increases the chance of bone diseases. Therefore, providing vitamin D supplements to breastfed infants in the first year of life appears appropriate.

## Figures and Tables

**Figure 1 nutrients-12-00404-f001:**
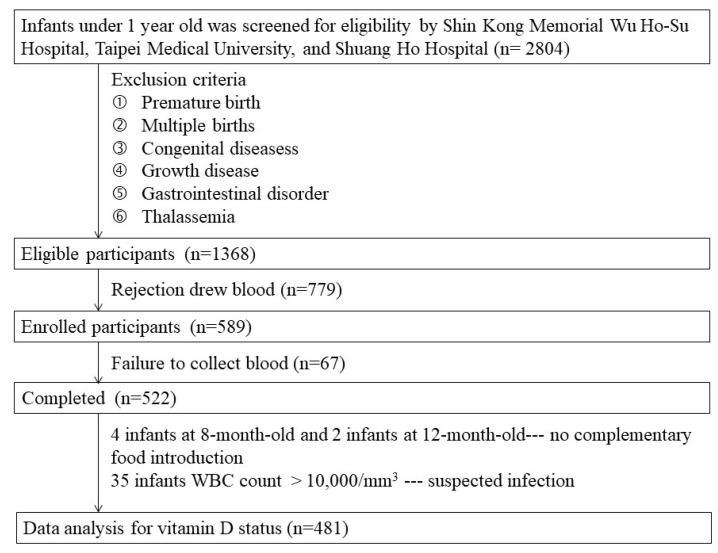
Flowchart of infant selection.

**Figure 2 nutrients-12-00404-f002:**
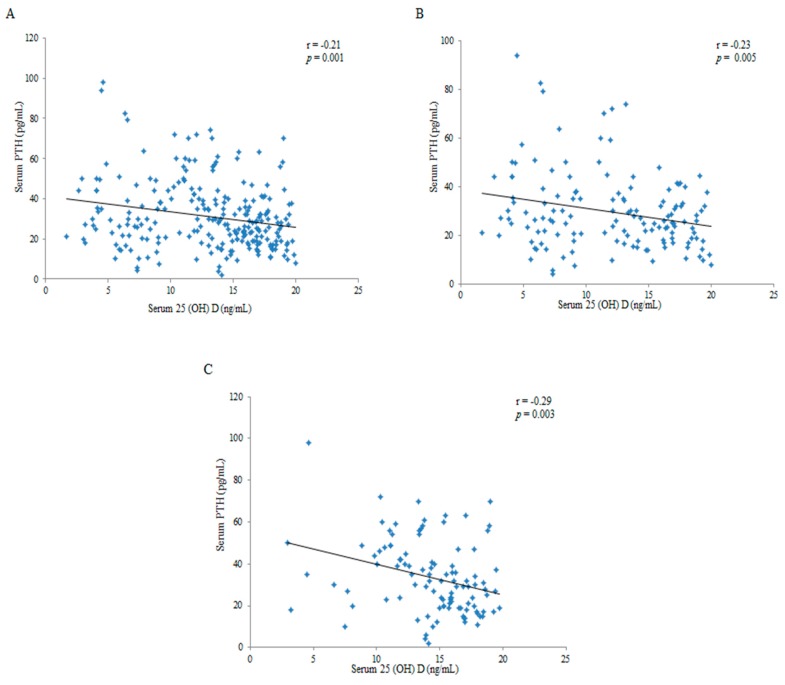
Correlations between serum 25(OH)D_3_ level and iPTH (intact parathyroid hormone) at 1–12 months (**A**), 1–6 months (**B**), and 7–12 months (**C**).

**Table 1 nutrients-12-00404-t001:** Demographic characteristics of infants diagnosed with vitamin D deficiency (VDD) or vitamin D insufficiency (VDI) in the first year of life ^a^.

	All	VDS ^b^	VDI	VDD	*p*-Value
Number (%)	481 (100)	72 (15.0)	197 (40.9)	212 (44.1)	
Sex					
Male (%)	270 (56.1)	34 (12.6)	106 (39.3)	130 (48.1)	0.19
Female (%)	211 (43.9)	38 (18.0)	91 (43.1)	82 (38.9)	
Chronological age (month)	6.5 ± 4.4	5.4 ± 3.1	6.2 ± 4.3	6.8 ± 4.7	0.08
Gestational age (week)	38.4 ± 1.4	38.1 ± 1.5	38.3 ± 1.4	38.5 ± 1.4	0.05
Birthweight (kg)	3.0 ± 0.5	2.9 ± 0.5	2.9 ± 0.5	3.0 ± 0.5	0.10
Feed type					
Breastfed	172 (35.8)	3 (1.7)	21 (12.2)	148 (86.1)	<0.001 *
Mix-fed	81 (16.8)	12 (14.8)	27(33.3)	42 (51.9)	
Formula-fed	228 (47.4)	48 (21.1)	92 (40.4)	88 (38.5)	
Exclusive breastfeeding (months)	3.8 ± 3.5	3.1 ± 3.3 ^a^	3.1 ± 3.1 ^a^	4.2 ± 3.6 ^b^	0.003 *
Birth season					
Summer and fall (%)	188 (39.1)	35 (18.6)	55 (29.3)	98 (52.1)	0.01 *
Winter and spring (%)	293 (60.9)	27 (9.2)	85 (29.0)	181 (61.8)	
Outdoor times weekly (times)	2.5 ± 3.2	2.3 ± 2.9	2.4 ± 3.2	2.6 ± 3.3	0.68

^a^ Values are expressed as mean ± SD or n (%). ^b^ According to the Endocrine Society guidelines, VDD, VDI, and vitamin D sufficiency (VDS) are 25(OH)D_3_ levels (ng/mL) ≤20, 21–29, and ≥30, respectively. * Differences between groups were tested using one-way ANOVA, followed by the Scheffé method for a post hoc or chi-square test; *p* < 0.05 was considered statistically significant.

**Table 2 nutrients-12-00404-t002:** Serum 25(OH)D_3_ concentrations and anthropometric in infants ^a^.

	All	VDS ^b^	VDI	VDD	*p*-Value
**Vitamin D status**					
Serum 25(OH)D_3_ (ng/mL)	19.7 ± 9.7	36.9 ± 7.8^a^	23.7 ± 2.7 ^b^	13.0 ± 4.7 ^c^	<0.001 *
Serum iPTH (pg/mL)	31 ± 18	30 ± 25	27 ± 21	31 ± 17	0.66
**Anthropometrics**					
Body weight percentile (%)	50 ± 29	49 ± 29	47 ± 28	53 ± 29	0.14
Body length percentile (%)	52 ± 31	53 ± 35	49 ± 29	55 ± 31	0.19
Head circumference (%tile)	56 ± 31	53 ± 30	54 ± 31	57 ± 30	0.46

^a^ Data are presented as mean ± SD. ^b^ VDD, vitamin D deficiency; VDI, vitamin D insufficiency; VDS, vitamin D sufficiency; and iPTH, intact parathyroid hormone. ^c^ Means in the column with different superscripts indicate significant difference (*p* < 0.05) tested with one-way ANOVA and followed by the Scheffé method for the post hoc test.

**Table 3 nutrients-12-00404-t003:** Vitamin D intake in infants aged 1–6 and 7–12 months ^a^.

	All	VDS ^b^	VDI	VDD	*p*-Value
**1–6 months**					
Number (%)	309 (100)	44 (14.2)	94 (30.4)	171 (55.4)	
Chronological age (months)	3.4 ± 1.8	3.8 ± 1.8	3.5 ± 1.8	3.3 ± 1.8	0.32
Breastfed (%)	184 (59.5)	10 (22.2)	37 (39.4)	137 (80.1)	<0.001 *
Vitamin D intake from formula (mg/d)	3.31 (6.67)	7.12 (4.91)	5.33 (5.29)	0.00 (4.67)	<0.001 *
Total vitamin D intake (mg/d) ^c^	3.66 (6.67)	8.33 (6.19)	5.33 (5.55)	0.82 (4.67)	<0.001 *
**7–12 months**					
Number (%)	172 (100)	19 (11.0)	47 (27.3)	106 (61.6)	
Chronological age (months)	11.3 ± 2.8	9.1 ± 2.2	11.2 ± 2.8	11.7 ± 2.7	<0.001 *
Breastfed (%)	57 (33.1)	5 (26.3)	6 (12.8)	46 (43.4)	<0.001 *
Vitamin D intake from formula (mg/d)	6.15 (6.16)	6.15 (6.06)	7.70 (2.39)	4.10 (6.84)	<0.001 *
Vitamin D intake from complementary food (μg/d)	1.51 (2.65)	1.58 (2.94)	1.40 (3.40)	1.53 (2.57)	0.81
Total vitamin D intake (mg/d)	7.67 (7.45)	9.41 (9.43)	9.34 (4.47)	5.96 (8.18)	<0.001 *

^a^ Data are presented as medians (interquartile range); ^b^ VDS, vitamin D sufficiency; VDI, vitamin D insufficiency; and VDD, vitamin D deficiency; and ^c^ Total vitamin D intake was the sum of vitamin D intake from formula milk and complementary food. * Intergroup differences were tested using chi-square test, one-way ANOVA, followed by the Scheffé method for a post hoc or the Kruskal–Wallis test; *p* < 0.05 was considered statistically significant.

**Table 4 nutrients-12-00404-t004:** Determinants of vitamin D deficiency among infants.

	Beta Coefficients	SE ^a^	OR	95% CI	*p*-Value
**Dietary Vitamin D Intake**					
Model 1 ^b^					
VDI	0.14	0.03	1.15	1.09–1.20	<0.001 *
VDD	0.21	0.03	1.23	1.16–1.31	<0.001 *
Model 2					
VDI	0.21	0.03	1.23	1.15–1.32	<0.001 *
VDD	0.34	0.05	1.40	1.28–1.53	<0.001 *
**The Period of Exclusive Breastfeeding**					
Model 1					
VDI	0.003	0.05	1.00	0.91–1.11	0.96
VDD	0.10	0.05	1.11	1.02–1.21	0.02 *
Model 2					
VDI	0.001	0.05	1.00	0.90–1.11	0.99
VDD	0.10	0.05	1.11	1.01–1.22	0.04 *
**Birth Season**					
Model 1					
VDI	0.70	0.31	2.02	1.10–3.7	0.02 *
VDD	0.86	0.29	2.37	1.35–4.17	0.003 *
Model 2					
VDI	0.83	0.32	2.30	1.22–4.34	0.01 *
VDD	0.98	0.31	2.66	1.46–4.86	0.001 *
**Outdoor Times Weekly**					
Model 1					
VDI	0.00	0.05	1.00	0.91–1.10	0.99
VDD	0.04	0.04	1.04	0.95–1.13	0.42
Model 2					
VDI	−0.01	0.05	0.99	0.89–1.09	0.79
VDD	0.02	0.05	1.02	0.93–1.12	0.62

^a^ SE^a^, standard error of mean; OR, odds ratio; 95% CI, 95% confidence interval; ^b^ Model 1: not adjusted; and Model 2: adjusted for gestational week, birthweight, sex, body weight percentile, body length percentile, and chronological age. * *p* < 0.05.

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
