# Peer review of "Infants’ Vitamin D Nutritional Status in the First Year of Life in Northern Taiwan"

_nutrients, 2020, doi:10.3390/nu12020404_

Round 1
Reviewer 1 Report
Thank you for this interesting article. It adds to the important body of evidence recommending that infants are given vitamin D supplementation.
It is very well written. I have just suggested a couple of corrections and clarification of units of measurement.
Author Response
Response to Reviewer 1 Comments
Point 1: ‘questionnaires were completed for 481 infants’
Response 1: Thank you for your pointing out. In L33, the sentence was updated ‘481 infants’ parents completed a questionnaire’
Point 2: I think you should just check the English here. Perhaps say leading to a poor source of vitamin D. ‘…vitamin D concentration cause to a poor source of vitamin D for the exclusive breast feeding’
Response 2: We agree with your opinion on this point. In L87, the sentence was updated ‘vitamin D concentration leading to a poor source of vitamin D for the exclusive breast feeding’
Reviewer 2 Report
Thank you for the opportunity to review this manuscript. The high prevalence of vitamin D insufficiency and deficiency worldwide and the high risk of deficiency in newborns and infants means this is an area of interest.
Abstract: Inclusion of the specific vitamin D cut-off concentrations in the abstract would be useful. Abstract: ‘Overall, 481 infants completed...’. Should this be parents/mothers completed the questionnaire? Abstract: It may be more appropriate to refer to anthropometrics (i.e. what has been measured – body weight, length and head circumference) rather than ‘growth’ when presenting vitamin D status and anthropometric data. Introduction: It may not be correct to state ‘most experts agree on the definitions from the Endocrine Society..’. One of the challenges of vitamin D research is that different insufficiency and deficiency thresholds have been proposed. For example, the Institute of Medicine guidelines for the USA and Canada suggest a 25(OH)D deficiency threshold of <30 nmol/L (12ng/mL) with an individual target of ≥50 nmol/L (20 ng/mL) (1). This should be considered in the discussion. Section 2.2. Given its importance to the manuscript, it may be useful to provide more information on the calculation of vitamin D intake from complementary foods. Section 2.2. ‘Vitamin D content in breast milk was minor and not evaluated’. Consider rephrasing to ‘The vitamin D content of breastmilk is reported to be minor and was not evaluated in this study’ and include a reference. Section 2.6: The vitamin D cut-offs described appear to be The Endocrine Society guidelines (2) rather than those of the American Association of Clinical Endocrinologists (3)? Section 3. The order and headings of results section are a little confusing, a slightly different order may be clearer for the reader. Consider; 3.1 Participant Characteristics - this should include a table of demographics (age, gender etc.) of the group as a whole (n = 481) 3.2 Vitamin D status – this should include the mean ± SD 25(OH)D concentration of the group as a whole, data on the relationship between 25(OH)D and PTH (currently in section 3.2) and differences in the prevalence of vitamin D insufficiency/deficiency across participant characteristics 3.3 Anthropometrics and vitamin D status 3.4 Vitamin D intake - this should include data on vitamin D intakes of the group as a whole Section 3.1. Where showing the difference in prevalence of vitamin D insufficiency/deficiency between groups e.g. with feeding methods, exclusive breastfeeding duration etc. in the text, please include P-values to show significance of results. Section 3.2. A correlation coefficient has been produced for PTH and 25(OH)D <20 ng/mL. Please provide data on the relationship between PTH and 25(OH)D >20 ng/mL. Section 3.4. This title is incorrect. If the aim of the logistic regression models was to identify predictors of vitamin D insufficiency and deficiency, was vitamin D intake the only significant predictor of vitamin D status? If not, as alluded to in the discussion, the association of each individual predictor with vitamin D status should be presented. Discussion: ‘The prevalence of vitamin D deficiency… and low in the United States and in the present study sample (44%)’. A prevalence of vitamin D deficiency of 44% cannot be considered to be low. Discussion:‘However, our study revealed that…but increased iPTH concentration in the first year of life’ The results do not back up this statement – although there was a negative correlation between PTH and 25(OH)D when 25(OH)D was <20ng/mL, there was no difference in PTH concentration between vitamin D sufficient and deficient groups. If referring to the small difference in correlation strength between PTH and 25(OH)D in infants aged 1-6 months and 7-12 months, consider separating these sentences for ease of understanding. Discussion: ‘Caregivers should be encouraged to increase the frequency of taking babies outside’. Consider the risk of skin damage from sun exposure. Recommendations, such as from the American Academy of Pediatrics, suggest avoiding direct, unprotected, sun exposure for newborns and infants. Discussion: It is noted that since 2016 the Taiwan Pediatric Association recommends a daily 400 IU supplement for fully and partially breastfed infants, which if adhered to, should improve the vitamin D status of breastfed infants. Was use of supplements examined in the current study? Supplements represent a potentially important source of vitamin D in infants.
Author Response
Response to Reviewer 2 Comments
Point 1: Abstract: Inclusion of the specific vitamin D cut-off concentrations in the abstract would be useful. Abstract: ‘Overall, 481 infants completed...’. Should this be parents/mothers completed the questionnaire?
Response 1: Thank you for your pointing out. In L32-33, we add ‘The specific vitamin D cut-off concentrations for VDD, VDI, and VDS are 25(OH)D3 level ≤ 20, 21–29, and ≥ 30 (ng/mL)’. In L33, the sentence was updated ‘481 infants’ parents completed a questionnaire’
Point 2: Abstract: It may be more appropriate to refer to anthropometrics (i.e. what has been measured – body weight, length and head circumference) rather than ‘growth’ when presenting vitamin D status and anthropometric data.
Response 2: We agree with your opinion on this point. In L41 and L89, the sentence was updated.
Point 3: Introduction: It may not be correct to state ‘most experts agree on the definitions from the Endocrine Society..’. One of the challenges of vitamin D research is that different insufficiency and deficiency thresholds have been proposed. For example, the Institute of Medicine guidelines for the USA and Canada suggest a 25(OH)D deficiency threshold of <30 nmol/L (12ng/mL) with an individual target of ≥50 nmol/L (20 ng/mL) (1). This should be considered in the discussion.
Response 3: Our authors agree with your keen opinion. We’ve added this point in the discussion (L289-298).
Point 4: Section 2.2. Given its importance to the manuscript, it may be useful to provide more information on the calculation of vitamin D intake from complementary foods.
Response 4: Thank you for your valid and constructive comments. We’ve added this point in section 2.2 (L125-131) and in our limitation (L393-395).
Point 5: Section 2.2. ‘Vitamin D content in breast milk was minor and not evaluated’. Consider rephrasing to ‘The vitamin D content of breastmilk is reported to be minor and was not evaluated in this study’ and include a reference.
Response 5: We appreciate your keen comments. In L132-133, the setence was updated.
Point 6: Section 2.6: The vitamin D cut-offs described appear to be The Endocrine Society guidelines (2) rather than those of the American Association of Clinical Endocrinologists (3)?
Response 6: In L181, the setence was updated.
Point 7: Section 3. The order and headings of results section are a little confusing, a slightly different order may be clearer for the reader. Consider; 3.1 Participant Characteristics - this should include a table of demographics (age, gender etc.) of the group as a whole (n = 481) 3.2 Vitamin D status – this should include the mean ± SD 25(OH)D concentration of the group as a whole, data on the relationship between 25(OH)D and PTH (currently in section 3.2) and differences in the prevalence of vitamin D insufficiency/deficiency across participant characteristics 3.3 Anthropometrics and vitamin D status 3.4 Vitamin D intake - this should include data on vitamin D intakes of the group as a whole Section 3.1. Where showing the difference in prevalence of vitamin D insufficiency/deficiency between groups e.g. with feeding methods, exclusive breastfeeding duration etc. in the text, please include P-values to show significance of results. Section 3.2. A correlation coefficient has been produced for PTH and 25(OH)D <20 ng/mL. Please provide data on the relationship between PTH and 25(OH)D >20 ng/mL. Section 3.4. This title is incorrect. If the aim of the logistic regression models was to identify predictors of vitamin D insufficiency and deficiency, was vitamin D intake the only significant predictor of vitamin D status? If not, as alluded to in the discussion, the association of each individual predictor with vitamin D status should be presented.
Response 7: Thank you for your valuable and constructive comments. The order and headings of results section are updated in results 3.1 (L194), 3.2 (L226-232), 3.3 (L244-248), 3.4 (L249-268), and 3.5 (L275-282). We also updated the table 1, table 2 and table 3 according reviewer’s suggestion.
Point 8: Discussion: ‘The prevalence of vitamin D deficiency… and low in the United States and in the present study sample (44%)’. A prevalence of vitamin D deficiency of 44% cannot be considered to be low.
Response 8: Thank you for your pointing out. In L300-302, the setence was updated.
Point 9: Discussion:‘However, our study revealed that…but increased iPTH concentration in the first year of life’ The results do not back up this statement – although there was a negative correlation between PTH and 25(OH)D when 25(OH)D was <20ng/mL, there was no difference in PTH concentration between vitamin D sufficient and deficient groups. If referring to the small difference in correlation strength between PTH and 25(OH)D in infants aged 1-6 months and 7-12 months, consider separating these sentences for ease of understanding.
Response 9: We agree with your opinion on this point. In L302-304, the setence was updated.
Point 10: Discussion: ‘Caregivers should be encouraged to increase the frequency of taking babies outside’. Consider the risk of skin damage from sun exposure. Recommendations, such as from the American Academy of Pediatrics, suggest avoiding direct, unprotected, sun exposure for newborns and infants..
Response 10: Thank you for your suggestions. We’ve added this point in L337-339.
Point 11: Discussion: It is noted that since 2016 the Taiwan Pediatric Association recommends a daily 400 IU supplement for fully and partially breastfed infants, which if adhered to, should improve the vitamin D status of breastfed infants. Was use of supplements examined in the current study? Supplements represent a potentially important source of vitamin D in infants
Response 11: Thank you for your pointing out. All of infants in present study were not suppled with vitamin D supplements. This study was supported by the Ministry of Health and Welfare in 2012. Before publish, we had provided these results to government and pediatric experts as reference for nutrition policymakers.
Reviewer 3 Report
General comments
In this epidemiological report the authors describe the vitamin D nutritional intake and status (measured by serum 25OHD) of 481 infants and toddlers. It is noteworthy that the percentages of infants with VDI and VDD are the same.
Specific comments
There are only a few comments mostly analytical. The authors describe the measurement metthods for iPTH and 25OHD. However they provide no information on their specificity, sensitivity, and uncertainty (intra- and inter-assay coefficients of variation). The authors should provide exemplary mass spectrometry profiles with ions selected for analysis.
They should also mention what internal standards were used (usually C13 stable isotopically labelled vitamin D metabolites. Furthermore, they go at great length describing the mass spectrometry method but fail to provide data on the capacity of identifying the 25OHD isomers/epimers. One of the strengths of the mass spectrometry approach is to be able do discern these molecules. The authors should provide data on the concentrations of 25OHD2, 25OHD3 & possibly on the respective C-3 epimers. The latter have been reported at higher concentration than in older children & adults.
The volume of blood sample taken seems high considering the age of the infants. The fact that an arterial sample was use is also ethically questionable. Arterial sampling are usually done for specific medical needs and under medical supervision.
The title for section 3.4 is inappropriate. It seems to come from another text. There is no mention of iron status and anaemia in the text.
Author Response
Response to Reviewer 3 Comments
Point 1: There are only a few comments mostly analytical. The authors describe the measurement metthods for iPTH and 25OHD. However they provide no information on their specificity, sensitivity, and uncertainty (intra- and inter-assay coefficients of variation). The authors should provide exemplary mass spectrometry profiles with ions selected for analysis.
Response 1: Thank you for your valid and constructive comments. We’ve added this point and some available information in the section 2.4 (L149-150) and section 2.5 (L173-179). We asked the technicians of Technology Commons in National Taiwan University for mass spectrometry profiles, they told us that data was lost due to machine failure many years ago. Sorry, we cannot provide the exemplary mass spectrometry profiles.
Point 2: They should also mention what internal standards were used (usually C13 stable isotopically labelled vitamin D metabolites. Furthermore, they go at great length describing the mass spectrometry method but fail to provide data on the capacity of identifying the 25OHD isomers/epimers. One of the strengths of the mass spectrometry approach is to be able do discern these molecules. The authors should provide data on the concentrations of 25OHD2, 25OHD3 & possibly on the respective C-3 epimers. The latter have been reported at higher concentration than in older children & adults.
Response 2: Our authors agree with your keen opinion. The vitamin D2 comes from plant-based foods, mostly from yeast and mushrooms. Since most infants did not use yeast and mushrooms as complementary food, all infants in present study were not suppled with vitamin D supplements, the oncentration of 25OHD2 can not be detected. So we revised the 25(OH)D to 25(OH)D3 in text and used 1α-hydroxyvitamin D3 as internal standard and added intra- and inter-assay CV% in section 2.5 (L153-154, L173-179). Although it has been reported that the concentration of C-3 epimers was higher in infants than in older children and adults, however the research technician told us they cannot be separated by our used chromatography columns. I am sorry can not provide these information.
Point 3: The volume of blood sample taken seems high considering the age of the infants. The fact that an arterial sample was use is also ethically questionable. Arterial sampling are usually done for specific medical needs and under medical supervision.
Response 3: Thank you for your pointing out. This study was part of a survey “The effects of breastfeeding on iron and vitamin D status in infants.” A total of 4 cc blood is needed for all biochemical tests. At the beginning, we started drawing blood by venous, but the blood flow was slow and coagulated easily, resulting in insufficient blood volume for analysis. So we switched to arterial puncture and it was easier to obtain sufficient blood volume. All infants were well cared by medical staff without comorbidities after the blood was drawn.
Point 4: The title for section 3.4 is inappropriate. It seems to come from another text. There is no mention of iron status and anaemia in the text.
Response 4: Thank you for your pointing out. In L275, section 3.5 was updated.
Round 2
Reviewer 3 Report
Comment
The authors have considerably modified and improved their manuscript. However there is a need for improvement in language, mainly sentence structure. Once these resolved the manuscript could be accepted for publication.
Section 2.2: Reference [22] should be placed right after lee et al., if this is meant.
Author Response
Response to Reviewer 3 Comments (Round 2)
Point 1: The authors have considerably modified and improved their manuscript. However there is a need for improvement in language, mainly sentence structure. Once these resolved the manuscript could be accepted for publication.
Response 1: Thank you for your valid and constructive comments. We had modified some sentence structure (L27-29, L37, L64-67, L102-103, L286-287, L381-84) and reduced the repetition rate of our article while revising initial manuscript by Turnitin software and DupliChecker website.
Point 2: Section 2.2: Reference [22] should be placed right after lee et al., if this is meant.
Response 2: Thank you for your pointing out. In L126, the reference [22] was placed in right site.